# Synthesis of Zn_x_Cd_1-x_Se@ZnO Hollow Spheres in Different Sizes for Quantum Dots Sensitized Solar Cells Application

**DOI:** 10.3390/nano9020132

**Published:** 2019-01-22

**Authors:** Libo Yu, Zhen Li

**Affiliations:** 1College of Chemistry and Chemical Engineering, Hexi University, Zhangye City 734000, Gansu Province, China; yulibo_665@163.com; 2Key Laboratory of Hexi Corridor Resources Utilization of Gansu, Hexi University, Zhangye City 734000, Gansu Province, China

**Keywords:** alloyed quantum dots, hollow spheres, sensitized solar cells, zinc oxide

## Abstract

Zn_x_Cd_1-x_Se@ZnO hollow spheres (HS) were successfully fabricated for application in quantum dot sensitized solar cells (QDSSCs) based on ZnO HS through the ion-exchange process. The sizes of the Zn_x_Cd_1-x_Se@ZnO HS could be tuned from ~300 nm to ~800 nm using ZnO HS pre-synthesized by different sizes of carbonaceous spheres as templates. The photovoltaic performance of QDSSCs, especially the short-circuit current density (J_sc_), experienced an obvious change when different sizes of Zn_x_Cd_1-x_Se@ZnO HS are employed. The Zn_x_Cd_1-x_Se@ZnO HS with an average size distribution of ~500 nm presented a better performance than the QDSSCs based on other sizes of Zn_x_Cd_1-x_Se@ZnO HS. When using the mixture of Zn_x_Cd_1-x_Se@ZnO HS with different sizes, the power conversion efficiency can be further improved. The size effect of the hollow spheres, light scattering, and composition gradient structure Zn_x_Cd_1-x_Se@ZnO HS are responsible for the enhancement of the photovoltaic performance.

## 1. Introduction

The huge consumption of fossil fuels, and global warming caused by worldwide industries, have forced the exploration of clean, environmentally friendly energy [1,2,3]. Solar energy is one of the best candidates for a future energy source. The common use of solar energy is for various photovoltaic devices including silicon solar cells, thin film solar cells, dye-sensitized solar cells, quantum dots sensitized solar cells, and perovskite solar cells [4,5,6]. Among them, quantum dot sensitized solar cells (QDSSCs) have become the focus of investigation, because of their intrinsic advantages, including their superior extinction coefficient, possibility of multiple excitons generation, and higher theoretical power conversion efficiency (44%) than semiconductor solar cells, based on the Schockley–Queisser limit (33%) [7,8,9]. The concept of QDSSCs originated from dye sensitized solar cells (DSSCs), and its configuration and working principle are also similar to DSSCs [10]. The DSSCs consists of three parts, including the photoanode, electrolyte, and counterelectrode, which have been well been developed in the past decade, especially in the building of different types of photoanodes and electrolytes [11,12,13]. Inspired by the development of DSSCs, the investigation of QDSSCs has also made great achievements in the past years.

The power conversion efficiency of QDSSCs has constantly improved in recent years as a result of research efforts. The most remarkable breakthrough is that a power conversion efficiency of over 10% was achieved using Zn–Cu–In–Se quantum dots based on a TiO_2_ film [14,15]. However, there are still many challenges to be overcome in order to reach the theoretical limit (44%), indicating that the design of QDSSCs has still not been well optimized. As a key part of the photoanode in QDSSCs, the metal oxide particles of ~25 nm are most commonly used because of a high specific surface area for QD loading [16,17,18]. However, typical metal oxide particles of ~25 nm in the photoanode are weak when generating light scattering, because the size of the particles is far smaller than the wavelength of the visible light, causing a less light harvesting efficiency. Based on the Mie theory and Anderson localization of light [19], the resonant scattering of light is predicted to occur for spherical particles only when the particle size is comparable to the wavelength of the incident light [20]. To tackle this issue, metal oxide hollow spheres seem to be an appropriate choice for application in QDSSCs. Among these hollow structures, ZnO hollow spheres (HS) are particularly attractive as photoanodes for QDSSCs, because of their high electron mobility and low production cost [21]. Moreover, the size of ZnO HS is easy to control during the fabrication process by using carbonaceous spheres as templates to generate better light scattering among hollow spheres, improving their light harvesting capability.

Using ZnO HS as a photoanode, many types of QDs, such as single CdS, CdSe, CdTe, PbS, and PbSe; cosensitized CdS/CdSe QDs; CdS_x_Se_1-x_; and Zn_x_Cd_1-x_Se alloyed QDs can be loaded for solar cell application [22,23,24,25,26,27]. Ternary alloyed QDs have drawn much attention recently because gradation composition heterostrucures can be formed in order to tailor the alignment of the conduction band edges of the QDs, which is helpful to improve the electron injection efficiency [28,29,30]. For example, ZnO–Zn_x_Cd_1-x_Se core/shell nanowire array QDSSCs form a stepwise energy alignment at the heterojunctions, where both the conduction and valence bands of the shell are either higher or lower in energy than that those of the core [24,28,29,30,31,32,33], leading to a preferable transfer of electrons across the interface from Zn_x_Cd_1-x_Se to ZnO. Although several types of QDSSCs based on ZnO nanostructures have been reported, it seems that the ternary alloyed QDs directly grown on ZnO hollow spheres have not been reported widely in QDSSCs’ application.

In view of these backgrounds, we constructed QDSSCs based on Zn_x_Cd_1-x_Se@ZnO hollow spheres (HS) with different sizes, by using a simple ion-exchange route. The strategy for the of fabrication of Zn_x_Cd_1-x_Se@ZnO HS is based on the difference of the solubility product constant (K_sp_) of ZnO (6.8 × 10^−17^), ZnSe (3.6 × 10^−26^), and CdSe (6.31 × 10^−36^) [33], implying that the pre-prepared ZnO HS can be employed as sacrificial templates to form Zn_x_Cd_1-x_Se@ZnO HS by Se^2−^ anion exchange and Cd^2+^ cation exchange in sequence. Furthermore, the size of the Zn_x_Cd_1-x_Se@ZnO HS could be controlled in accordance with the visible light region by using carbonaceous spherical templates, contributing to a stronger light scattering ability. To our best knowledge, the investigation of the size effect of Zn_x_Cd_1-x_Se@ZnO HS on the photovoltaic performance of QDSSCs has seldom been reported. Based on experimental results, the reasons for the enhancement of the photovoltaic performance of Zn_x_Cd_1-x_Se@ZnO HS QDSSCs are discussed.

## 2. Materials and Methods

### 2.1. Materials

The commercial chemical reagents, including sucrose, zinc nitrate, cadmium nitrate, selenium powder, sodium borohydride, ethylcellulose, terpinol, ethanol, sodium sulfide, and sulfur powder, were obtained from Aladdin Co. Ltd (Shanghai, China). The fluorine-doped tin oxide (FTO) conductive glass was purchased from Opvtech Co. Ltd (Yingkou, China). All of the materials were used directly without further purification.

### 2.2. Preparation of ZnO Hollow Microspheres

The ZnO hollow spheres (HS) were synthesized using a carbonaceous microspheres template method [34]. In a typical synthesis route, the carbonaceous microspheres, which were obtained by a hydrothermal process of a sucrose aqueous solution in a Teflon-stainless autoclave at 180 °C for 8 h (the size of carbonaceous can be modulated with different concentration of sucrose solution), were dispersed in the 1 M aqueous solution of zinc nitrate under ultrasonic for 20 min. Then, the suspension was aged for 6 h. After aging, the suspension was filtered, washed, and dried in order to get black powders. Subsequently, the black powders were heated to 500 °C in a muffle furnace at a rate of 2 °C·min^−1^, with holding of the temperature at 500 °C for 1 h. Finally, the resultant ZnO HS powders in white were acquired. The size of the ZnO HS can be controlled using carbonaceous microspheres with different sizes.

### 2.3. Construction of Zn_x_Cd_1-x_Se@ZnO HS Photoanodes and QDSSCs

The ZnO HS powders (3 g), ethylcellulose (0.5 g), terpinol (10 mL), and ethanol (3 mL), were mixed together under magnetic stirring in order to form a viscous paste. The ZnO paste was doctor-bladed onto the FTO glass (2.0 × 1.5 cm). The ZnO film active area was controlled to be 0.25 cm^2^, and the thickness of the film was tuned to be ~15 μm, using the same thickness spacers. After drying in ambient conditions, the products were annealed in a muffle furnace at 500 °C for 1 h in order to eliminate the organic residuals. The types of photoanodes constructed by different sizes of ZnO HS, including 300, 400, 500, and 800 nm, as well as a mixture of ZnO HS with different sizes (25 wt % for each size ZnO HS), were prepared for comparative investigation. 

Zn_x_Cd_1-x_Se@ZnO was obtained by immersing ZnO HS photoanodes in Se^2−^ and Cd^2+^ aqueous solutions, respectively. Firstly, the ZnO HS were immersed in a 0.1 M Se^2−^ ion solution prepared by reacting the Se powder with NaBH_4_. The immersing process was kept at 80 °C for 12 h, and was repeated two times in order to get a desirable ZnSe thickness of ZnSe@ZnO HS. Then, the ZnSe@ZnO HS were put into a 0.1 M Cd^2+^ solution at 80 °C for 12 h; this process was also repeated two times in order to gain the final product of Zn_x_Cd_1-x_Se@ZnO HS photoanodes.

For the solar cells application, the Zn_x_Cd_1-x_Se@ZnO HS photoanode and Cu_2_S counter electrode, which was prepared according to the previous literature [35], were assembled together by filling one drop of electrolyte consisting of 1 M S and 1 M Na_2_S, in a water/methanol (1:1 in volume ratio) solution. We prepared the QDSSCs based on Zn_x_Cd_1-x_Se@ZnO HS of different sizes in order for a comparative investigation of the size effect on the performance of the QDSSCs, and, moreover, three photoanodes based on a mixture of ZnO HS with different sizes (25 wt % for each size ZnO HS) were prepared under the same conditions for further investigation of the *I–V* performance’s reproducibility.

### 2.4. Characterization

We employed a Quanta 450 FEG scanning electron microscopy (SEM, Hillsboro, OR, USA) and Tecnai G2 F20 transmission electron microscope (TEM, Hillsboro, TX, USA) equipped with an energy dispersive X-ray spectrometer (EDS) for elemental analysis in order to record the morphology of the prepared products. The optical absorption properties of the photoanodes were recorded using a U-3900H UV-VIS spectrophotometer (Dallas, TX, USA), which was equipped with an integrating sphere attachment for diffuse reflection measurement.

With the assistant of the Oriel *I*–*V* test station, we investigated the *I*–*V* performance of the QDSSCs. A solar simulator was used to simulate sunlight illumination with an intensity of 100 mW·cm^−2^. The incident photon to the charge carrier generation efficiency (IPCE) was measured as a function of wavelength, using a 150 W Xe lamp coupled with a computer controlled monochromator.

## 3. Results and Discussions

The carbonaceous spheres were important for the synthesis of ZnO HS, because of their role as the template. The size controllable preparation of the carbonaceous spheres can be completed by varying the concentration of the sucrose aqueous solution. Figure 1 records the morphology changes of the resultant carbonaceous spheres as there is an increase in the sucrose concentration. The diameters of these carbonaceous spheres could be tuned from ~400 nm to ~2 μm, by simply increasing the sucrose concentration from 0.5 M to 2 M. Based on these experimental facts, we selected the carbonaceous spheres in sizes of ~400, ~ 600, and ~ 800 nm, and 1 μm, as templates. Because the scattering of light by spherical particles may occur when the particle size is comparable to the wavelength of the incident light according to the Mie theory [19], these selected carbonaceous spheres are available for the synthesis of ZnO HS in an appropriate diameter in order to generate light scattering.

ZnO HS can be obtained using carbonaceous spheres as a template. Figure 2a shows the SEM image of the carbonaceous sphere with a size of ~600 nm; from one of the broken spheres, it can be seen that they are solid spheres. After the formation of ZnO HS, the surface morphology of the spheres changed to being rougher than the original carbonaceous spheres. As shown in Figure 2b, it seems that the ZnO HS is aggregated by a large number of nanoparticles. The average diameter of ZnO HS is ~500 nm, which is smaller than that of the carbonaceous sphere templates, demonstrating that a shrinkage of ZnO HS occurred during the heating process. This phenomenon is in accordance with previous reports [36,37]. The inset of Figure 2b provides an SEM image of a broken ZnO HS, the empty inside can be easily discerned, proved the obtained spherical structure is of hollow spheres.

The hollow structures of the ZnO HS were further confirmed by TEM analysis, which is presented in Figure 3a–d. Generally, all of the products show a hollow spherical structure with an identifiable shell, indicating that ZnO HS can be successfully synthesized using carbonaceous spheres as templates. Moreover, the TEM results also indicate that the size of the ZnO HS can be controlled using carbonaceous spheres with different sizes. As shown in Figure 3a–d, the sizes of the ZnO HS can be tuned from ~300 nm to ~800 nm, using different sizes of the carbonaceous templates. Figure 3e is a selective area electron diffraction (SAED) pattern of ZnO HS. The appearance of the diffraction rings indicates that the ZnO HS are in polycrystalline structure. The lattice fringes of the ZnO HS are observed using the HRTEM image in Figure 3f. By careful measurements, the lattice spacing is measured to 0.26 nm, which corresponds to the (002) plane of hexagonal ZnO (JCPDS # 36-1451).

Based on the ZnO HS, the Zn_x_Cd_1-x_Se@ZnO HS were further fabricated through an ion-exchange process by soaking the ZnO HS in Se^2−^ and Cd^2+^ aqueous solutions, respectively. The morphology variation was recorded using SEM and TEM images. Figure 4a and its inset show the surface morphology of ZnO HS, and a large particles structure can be observed. The fine structure is further revealed by the TEM in Figure 4b, and the obvious shell and empty inside confirm the hollow spherical structure. The composition of the hollow sphere is shown in Figure 4c; except for the Cu element from the carbon film supported by the copper grid, only the Zn and O elements appear, and the atomic ratio is close to 1:1, confirming the formation of ZnO HS. After the ion-exchange process, a slight morphology variation on the surface of the hollow sphere is shown in Figure 4d and its inset. Smaller nanoparticles are formed on surface, indicating that a new substance has been formed on the surface of ZnO HS. In order to testify that the Zn_x_Cd_1-x_Se can be formed on the surface of ZnO HS, the elemental mapping test results are shown in Appendix A. As is shown, the Zn, Cd, Se, and O elements can be scanned on the surface of the hollow sphere, indicating that Zn_x_Cd_1-x_Se has formed on the surface of ZnO after the ion-exchange process. Moreover, the TEM image in Figure 4e shows an evident morphological variation after the ion-exchange process, also implying that some new materials formed on surface of the ZnO HS. In addition, the EDS analysis of Figure 4f from a designated inner spot of the hollow sphere shell shows that after the ion-exchange process, Cd and Se elements can also be found on the inner shell, and the atomic ratio of (Zn and Cd) to (Se and O) is close to 1:1, indicating the successful formation of Zn_x_Cd_1-x_Se@ZnO HS.

Based on the experimental facts and TEM results, the formation mechanism of the Zn_x_Cd_1-x_Se@ZnO HS is proposed in Figure 5. The first important step is the use of carbonaceous spheres rich with surface carboxyl and hydroxyl functional groups, which are affinity to the Zn^2+^ ion [38] for a large amount of ion adsorption. The carbonaceous spheres are immersed in a 1 M Zn^2+^ aqueous solution for 6 h, and then taken out to be heated in air. During sintering, the carbonaceous spheres turn to CO_2_, leading to the formation of ZnO hollow spheres. The size of the ZnO hollow spheres are tuned by selecting different size of carbonaceous sphere templates. The second key step is the ion-exchange process in order to form Zn_x_Cd_1-x_Se@ZnO HS. A sharp difference in the solubility product constant (K_sp_) among ZnO (6.8 × 10^−17^), ZnSe (3.6 × 10^−26^), and CdSe (6.1 × 10^−36^) [28,33] makes it possible for the ZnO HS to be used as sacrificial templates in order to get more stable ZnSe@ZnO HS and to further be converted into Zn_x_Cd_1-x_Se@ZnO HS. As illustrated in Figure 5, the ZnO hollow spheres initially conducted an anion exchange with Se^2−^ ions to form ZnSe@ZnO hollow spheres, and followed with a surface conversion of ZnSe to Zn_x_Cd_1-x_Se through the cation replacement of Zn^2+^ by Cd^2+^, leading to the final Zn_x_Cd_1-x_Se@ZnO hollow spheres (HS).

The UV-VIS absorption spectra of the photoanodes are illustrated in Figure 6. The absorption onset of the ZnO HS centers is ~385 nm, corresponding to the band gap of 3.2 eV. After the anion exchange process, the absorption onset exhibits an obvious redshift to ~460 nm, corresponding to the band gap of 2.60 eV. This redshift phenomenon indicates that the formation of ZnSe@ZnO HS enlarges the light absorption range to the visible region. A further redshift of the absorption onset to 690 nm, which corresponds to 1.79 eV, implies the successful formation of Zn_x_Cd_1-x_Se@ZnO HS after the cation exchange process. The significant enlargement of the light absorption means that most of the light in the visible region can be involved in order to excite Zn_x_Cd_1-x_Se for the generation of electrons, providing potential application in QDSSCs.

As the size of the Zn_x_Cd_1-x_Se@ZnO HS can be controlled using carbonaceous spheres with different sizes, it is reasonable to believe that the Zn_x_Cd_1-x_Se@ZnO HS photoanodes will have effects on the photovoltaic performance in the QDSSCs. Figure 7a records the current density–voltage (J–V) behavior of the QDSSCs assembled with Zn_x_Cd_1-x_Se@ZnO HS. The Zn_x_Cd_1-x_Se@ZnO hollow spheres with sizes of ~300, ~400, ~500, and ~800 nm are revealed by the TEM image, as shown in Figure 7b–e. The corresponding photovoltaic parameters of these QDSSCs including short-circuit current density (J_sc_), open-circuit voltage (V_oc_), fill factor (FF), and power conversion efficiency (PCE), are also summarized in Table 1.

The QDSSC based on the ~300 nm Zn_x_Cd_1-x_Se@ZnO HS photoanode exhibit a J_sc_ of 11.30 mA·cm^−2^, a V_oc_ of 0.42 V, and a FF of 0.44, producing a PCE of 2.07%. As the size of the Zn_x_Cd_1-x_Se@ZnO HS increases to ~400 nm, the QDSSC shows a J_sc_ of 11.61 mA·cm^−2^, a V_oc_ of 0.41 V, and a FF of 0.46, leading to an increase in PCE to 2.19%. A higher PCE of QDSSC is acquired by ~500 nm Zn_x_Cd_1-x_Se@ZnO HS, which presents a J_sc_ of 13.46 mA·cm^−2^, a V_oc_ of 0.42 V, a FF of 0.46, yielding the highest PCE of 2.60% among these sample QDSSCs. However, further increasing the Zn_x_Cd_1-x_S@ZnO HS to ~800 nm leads to a decrease of PCE to 1.72%, with J_sc_ of 8.79 mA·cm^−2^, V_oc_ of 0.40 V, and FF of 0.49. Apparently, the only difference among these sample QDSSCs is the size of the Zn_x_Cd_1-x_Se@ZnO HS. Therefore, we believe that a size effect of the Zn_x_Cd_1-x_Se@ZnO hollow spheres would probably be responsible for the changes of the photovoltaic performance. 

The explanations for the photovoltaic performance variation related to the structure of the Zn_x_Cd_1-x_Se@ZnO HS can be illustrated in Figure 8. According to the J–V curves results (Figure 7), it can be concluded that the change of J_sc_ among these Zn_x_Cd_1-x_Se@ZnO solar cells is the main reason for the difference in the photovoltaic performance. We believe that three factors, including light scattering on the surface of the hollow spheres, light reflection in the hollow spheres, and a band edges realignment caused by the gradient Zn_x_Cd_1-x_Se@ZnO structure, have influence on the variation of J_sc_. Figure 8a gives the working principle of QDSSC based on the Zn_x_Cd_1-x_Se@ZnO photoanode. Under illumination, the Zn_x_Cd_1-x_Se@ZnO will be excited so as to generate electrons. More light being absorbed will produce more electrons favorable to the increase of J_sc_. However, a significant portion of the light emitted on the photoanodes would transmit through the photoanode, without interacting with the Zn_x_Cd_1-x_Se@ZnO HS because of the smaller size hollow spheres. The resonant scattering of light is anticipated to occur for spherical particles in a size comparable to the incident light, according to the Mie theory and the Anderson localization of light [39]. In our case, the photoanode is packed by the Zn_x_Cd_1-x_Se@ZnO HS, by controlling the size of the hollow spheres, ~500 nm Zn_x_Cd_1-x_Se@ZnO HS is in the range of visible light, which results in a stronger scattering effect among the Zn_x_Cd_1-x_Se@ZnO hollow spheres than other sized hollow spheres, providing the photons more chances to be absorbed, and eventually leading to an enhanced light harvesting efficiency. In addition, the hollow spheres structure of Zn_x_Cd_1-x_Se@ZnO allows for light to be reflected by many times when it encounters the shell of the hollow sphere, because of its curved surface, as shown in Figure 8b, also leading to the enhancement of the light harvesting efficiency.

A sandwich composition gradient structure of Zn_x_Cd_1-x_Se@ZnO would be formed on the shell of the hollow spheres because of the the ion-exchange process, as illustrated in Figure 8c. The ZnSe layer is firstly formed on the ZnO HS surface during the Se^2−^ ion-exchange process, then, a Cd^2+^ ion-exchange process replaced the Zn^2+^ from the surface of the ZnSe layer, forming a sandwich composition gradient structure of Zn_x_Cd_1-x_Se@ZnO HS with a Cd^2+^ rich in surface [22,33]. This sandwich structure caused a band realignment, as shown in Figure 8d. When the Zn_x_Cd_1-x_Se@ZnO formed, the ZnO, ZnSe, and Zn_x_Cd_1-x_Se were brought into contact closely, and the energy level difference between ZnSe and Zn_x_Cd_1-x_Se would cause an electron flow from ZnSe to Zn_x_Cd_1-x_Se, which is known as the Fermi level alignment, triggering a downward and upward shift of band edges of ZnSe and Zn_x_Cd_1-x_Se, respectively [32]. The formation of a stepwise conduction band edge alignment in the order of Zn_x_Cd_1-x_Se > ZnSe > ZnO for a sandwich composition gradient Zn_x_Cd_1-x_Se@ZnO HS is beneficial for enhancing the injection and collection of photoexcited electrons to the conduction band of the ZnO layer.

The IPCE spectra of the QDSSCs based on the Zn_x_Cd_1-x_Se@ZnO HS with different sizes can provide more information on the enhancement of J_sc_. As seen in Figure 9, two interesting observations can be identified when comparing these spectra. The first one is that the range of the photoresponse centers is in the visible light region for all of the sample solar cells, demonstrating the excitation of Zn_x_Cd_1-x_Se as the primary event responsible for photocurrent generation. The second is a considerable increase of IPCE with the increasing size of Zn_x_Cd_1-x_Se@ZnO HS in the visible light region, especially the highest IPCE value produced by ~500 nm Zn_x_Cd_1-x_Se@ZnO HS. Considering its size in the range of the visible light region, its IPCE enhancement is mainly attributed to the increase of light scattering, contributing eventually to the improvement of J_sc_.

According to the above discussion, a ~ 500 nm Zn_x_Cd_1-x_Se@ZnO HS shows better scattering of light in the visible light region than the other HS sizes. However, the single size distribution of hte hollow spheres seems to not take full use of the visible light region in order to generate light scattering. Therefore, the photoanode constructed by a mixture of ZnO HS with different sizes from 300 to 800 nm was fabricated, and its photovoltaic performance is shown in Figure 10. As the J–V curves are shown in Figure 10a, and the mixture of the hollow spheres with different sizes can further improve the J_sc_ to 20.77 mA·cm^−2^, leading to the enhancement of PCE to 2.95%. In order to guarantee the performance’s reproducibility of our QDSSC based on ZnO HS with different sizes from 300 nm to 800 nm, we also prepared three QDSSCs under the same conditions for the J–V test, in order to investigate the stability of the photoanodes. The compared J–V test results are shown in Appendix A, which shows a relative stable variation PCE performance (from 3.05% to 2.92%). In comparison with a previous report, our QDSSC’s J_sc_ and PCE outperformed the similar ZnO-Zn_x_Cd_1-x_Se QDSSC based on ZnO nanowire arrays (J_sc_ = 6.7 mA·cm^−2^, V_oc_ = 0.64 V, FF = 0.35, PCE = 1.5%) [32], indicating this kind of structure’s potential advantages in improving the J_sc_ of solar cells. The IPCE in Figure 10b shows that the maximum IPCE value can reach to 85%, indicating a better utilization of the visible light region in order to enhance the light harvesting efficiency. This significant improvement of the photovoltaic performance using a mixture of hollow spheres can be ascribed to the enhanced light scattering in the different wavelengths of light caused by the wide range of sizes, from 300 to 800 nm, as shown in Figure 10c, compared with the single size of the hollow spheres, indicating the potential application of this strategy for designing QDSSCs.

## 4. Conclusions

In summary, unique Zn_x_Cd_1-x_Se@ZnO hollow spheres (HS) with a gradient position are successfully fabricated by an ion-exchange process with assistance from carbonaceous spheres as the template. The size of the Zn_x_Cd_1-x_Se@ZnO hollow spheres can be controlled by using carbonaceous spheres of different sizes. The influence on QDSSSs caused by the size variations of the Zn_x_Cd_1-x_Se@ZnO hollow spheres was investigated. Light scattering and the composition gradient structure of Zn_x_Cd_1-x_Se@ZnO hollow spheres are responsible for the enhancement of the photovoltaic performance. The size of the Zn_x_Cd_1-x_Se@ZnO hollow spheres at ~500 nm shows a better effect than the other sizes of the hollow spheres. By using a mixture of hollow spheres, with different sizes from 300 nm to 800 nm, the PCE of the QDSSC can be improved to 2.95%, showing the feasibility of the Zn_x_Cd_1-x_Se@ZnO hollow spheres in the design of high efficient QDSSCs.

## Figures and Tables

**Figure 1 nanomaterials-09-00132-f001:**
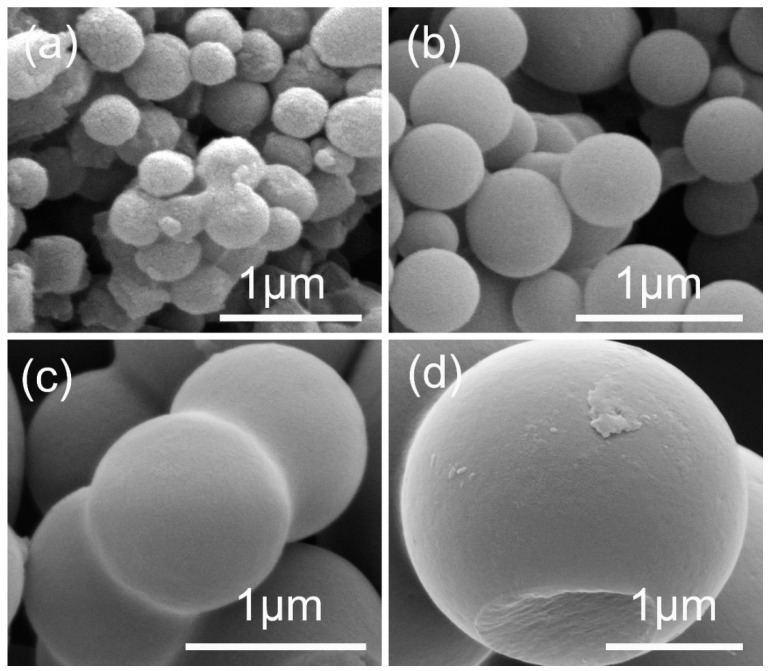
SEM images of carbonaceous spheres of different size prepared by varying the concentration of sucrose: (**a**) 0.5 M; (**b**) 0.75 M; (**c**) 1 M; (**d**) 2 M.

**Figure 2 nanomaterials-09-00132-f002:**
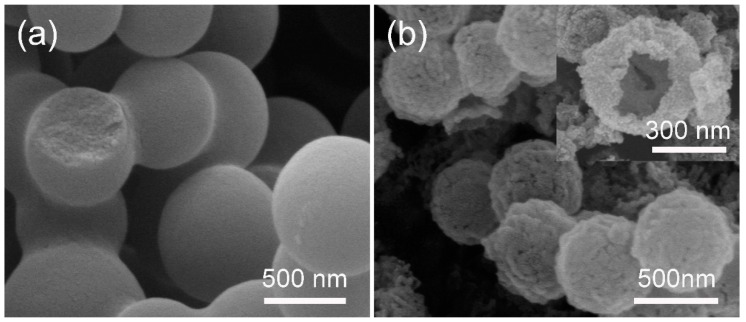
(**a**) SEM image of carbonaceous spheres template with size of ~600 nm; (**b**) ZnO hollow spheres (HS) obtained by soaking carbonaceous spheres in a 1 M zinc nitrate solution; the inset is the SEM of a broken ZnO HS.

**Figure 3 nanomaterials-09-00132-f003:**
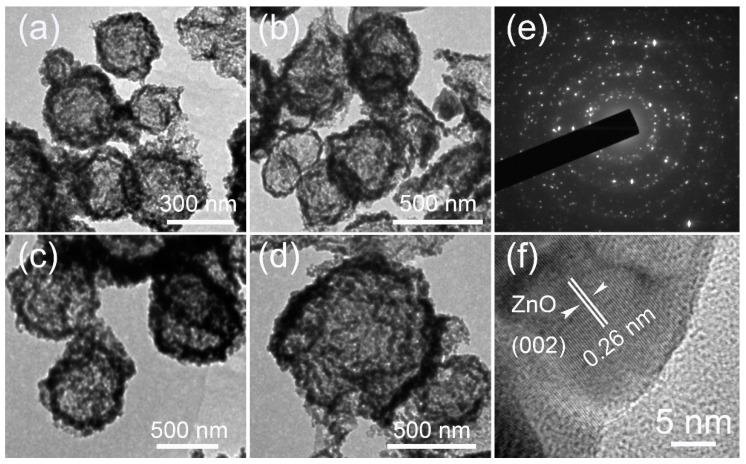
(**a**–**d**) TEM images of ZnO HS with a size of ~300, ~400, ~500 nm, and ~800 nm, respectively; (**e**) SAED pattern of ZnO HS; (**f**) the HRTEM of ZnO HS.

**Figure 4 nanomaterials-09-00132-f004:**
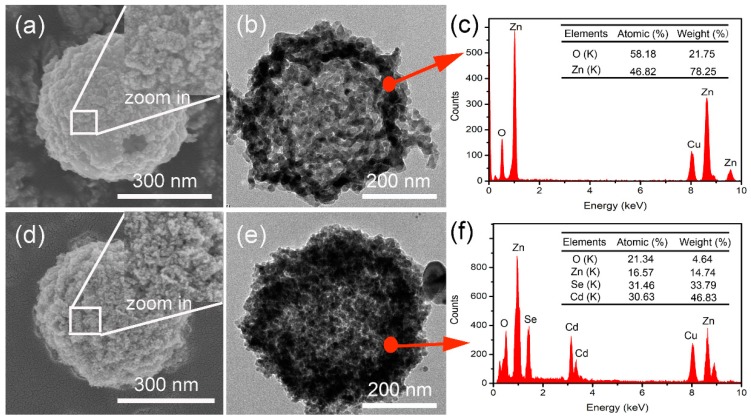
(**a**) SEM of ZnO HS, the inset is the magnified image of the selected zone; (**b**) TEM of ZnO HS; (**c**) energy dispersive X-ray spectrometer (EDS) of ZnO HS; (**d**) SEM of Zn_x_Cd_1-x_Se@ZnO HS, the inset is the magnified image of the selected zone; (**e**) TEM of Zn_x_Cd_1-x_Se@ZnO HS; (**f**) EDS of Zn_x_Cd_1-x_Se@ZnO HS.

**Figure 5 nanomaterials-09-00132-f005:**
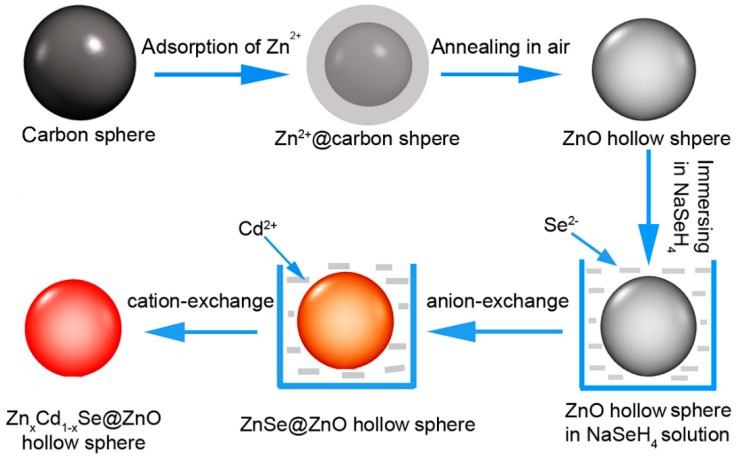
Illustration of the template approach to fabrication of ZnO HS and the ion-exchange process to the preparation of Zn_x_Cd_1-x_Se@ZnO HS.

**Figure 6 nanomaterials-09-00132-f006:**
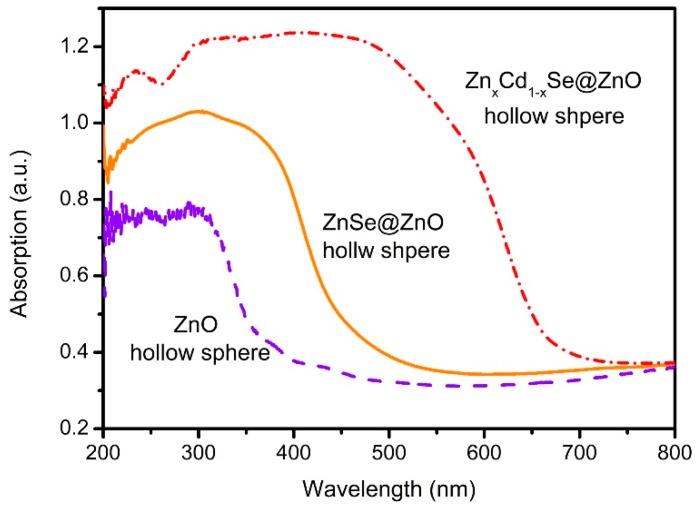
UV-visible absorption spectra of ZnO HS, ZnSe@ZnO HS, and Zn_x_Cd_1-x_Se@ZnO HS photoanodes, respectively.

**Figure 7 nanomaterials-09-00132-f007:**
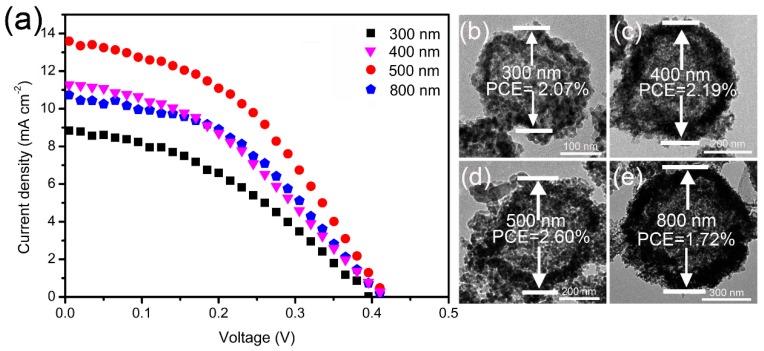
(**a**) Current density–voltage (J–V) curves of quantum dot sensitized solar cells (QDSSCs) based on Zn_x_Cd_1-x_Se@ZnO HS with a size of 300, 400, 500, and 800 nm; (**b**–**e**) the corresponding TEM images of Zn_x_Cd_1-x_Se@ZnO HS in different sizes with their power conversion efficiency (PCE).

**Figure 8 nanomaterials-09-00132-f008:**
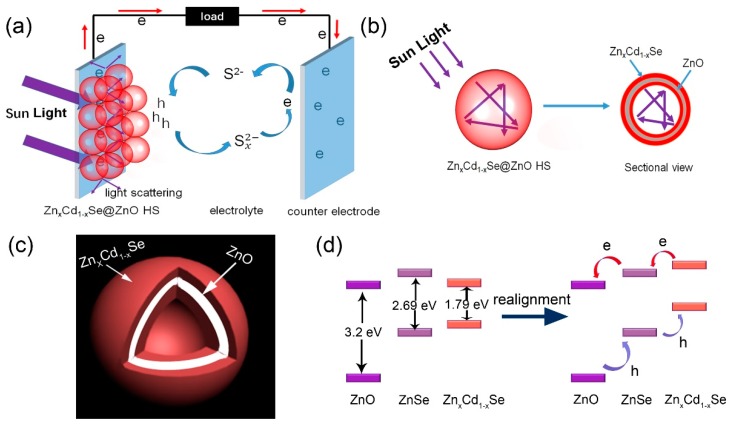
(**a**) The model of the QDSSC based on the Zn_x_Cd_1-x_Se@ZnO photoanode; (**b**) the light pathway inside the Zn_x_Cd_1-x_Se@ZnO HS; (**c**) the composition gradient structural model of Zn_x_Cd_1-x_Se@ZnO HS; (**d**) band edges realignment in Zn_x_Cd_1-x_Se@ZnO HS.

**Figure 9 nanomaterials-09-00132-f009:**
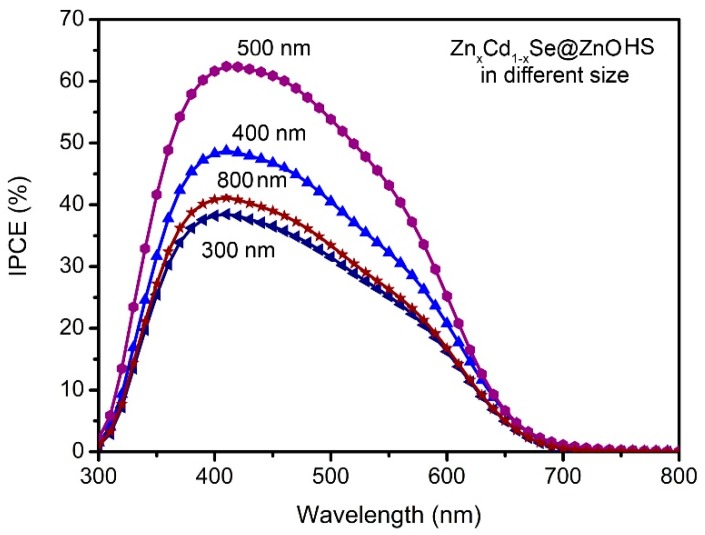
The incident photo-to-current conversion efficiency (IPCE) spectra of QDSSCs based on Zn_x_Cd_1-x_Se@ZnO HS in the sizes of ~300, ~400, ~500, and ~800 nm, respectively.

**Figure 10 nanomaterials-09-00132-f010:**
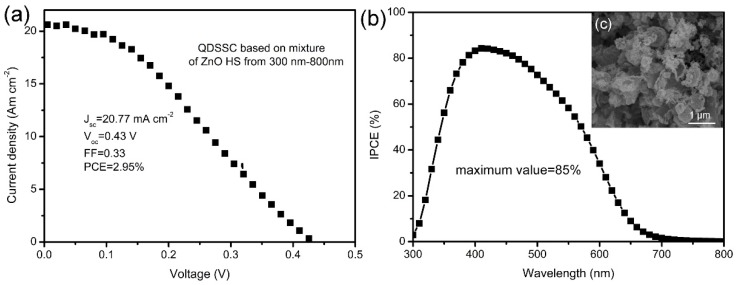
(**a**) The J–V curve of QDSSC based on a mixture of ZnO HS in different sizes, from 300–800 nm; (**b**) the corresponding IPCE spectrum; (**c**) the SEM image of the corresponding photoanode.

**Table 1 nanomaterials-09-00132-t001:** Size influence of Zn_x_Cd_1-x_Se@ZnO hollow spheres (HS) on photovoltaic performance. FF—fill factor; PCE—power conversion efficiency.

Size (nm)	J_sc_ (mA·cm^−2^)	V_oc_ (V)	FF	PCE (%)
~300	11.30	0.42	0.44	2.07
~400	11.61	0.41	0.46	2.19
~500	13.46	0.42	0.46	2.60
~800	8.79	0.40	0.49	1.72

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
