# Peer review of "Synthesis of ZnxCd1-xSe@ZnO Hollow Spheres in Different Sizes for Quantum Dots Sensitized Solar Cells Application"

_nanomaterials, 2019, doi:10.3390/nano9020132_

Reviewer 1 Report

The manuscript by Yu and Li describes the preparation of submicrometer-sized Zn-Cd-Se hollow spheres (HS) and the use of these in solar cells. In order to make the Zn-Cd-Se HS, the authors use a series of processing steps, including templating of solid carbon spheres to form ZnO HS and then step-wise ion exchange to (at least partially) convert the ZnO to Zn-Cd-Se HS. The micrometer size of the spheres improves the light scattering in the visible light range and in that way also the performance of the solar cells. In general, I think the study is well executed and worth publishing. However, few things should be clarified before the paper can be published:

1.       It is a bit unclear which parts of the processing steps are new (and which have been published before). For instance, a lot of focus is put on making the carbon spheres and the ZnO HS, although I believe these have been reported before. This should be clearly stated with proper referencing.

2.       The manuscript contains a lot of mistakes and the English is not that good. Please carefully go through the text and figures again.

3.       The authors claim that the presented solar cells are QD-sensitized solar cells (QDSSC). However, to me it is unclear if these cells can really be called QDSSC. This would require that the Zn-Cd-Se material consisted of nanometer-sized particles (which it might do). Figure 6 clearly highlights that the conversion affects the band gap of the HS. However, is it really a quantum effect or would you expect the same thing when converting bulk ZnO to bulk Zn-Cd-Se? Nonetheless, in the absence of a better name for this type of solar cell, QDSSC would be fine (although it should be discussed).

4.       Step-wise ion exchange reactions are used to convert the ZnO to Zn-Cd-Se. The authors claim that this produces a double shell structure (ZnxCd1-xSe@ZnO). However, in my opinion, there is no evidence for this (e.g. EDS is not sensitive enough to claim this). I see no reason why not the entire ZnO structure would be converted to Zn-Cd-Se. Has this been tested by for instance a reaction time series? Furthermore, could the smaller spheres be more efficiently converted to Zn-Cd-Se compared to the large ones?

5.       The authors claim that there are small particles inside the HS in the TEM image in Figure 4e. This is not necessarily true as the darker regions can also be from the top and bottom of the particle.

6.       The authors make a long argument about how the “gradient structure” will improve the performance of the devices due to band edge realignment. In addition to the lack of evidence for a gradient structure (see point 4), there is also no evidence to claim this. Furthermore, according to the model described by the authors, the ZnO seems to be isolated inside the HS. This means that there would be no pathway of electrons throughout the photoanode. Thus, I think the argument in Figure 8d is quite much speculation (and perhaps not necessary).   

7.       How do the current devices perform in comparison to other related devices in the literature?

Author Response

Respond to Reviewer #1

Comments and Suggestions for Authors

The manuscript by Yu and Li describes the preparation of submicrometer-sized Zn-Cd-Se hollow spheres (HS) and the use of these in solar cells. In order to make the Zn-Cd-Se HS, the authors use a series of processing steps, including templating of solid carbon spheres to form ZnO HS and then step-wise ion exchange to (at least partially) convert the ZnO to Zn-Cd-Se HS. The micrometer size of the spheres improves the light scattering in the visible light range and in that way also the performance of the solar cells. In general, I think the study is well executed and worth publishing. However, few things should be clarified before the paper can be published:

Comment 1.       It is a bit unclear which parts of the processing steps are new (and which have been published before). For instance, a lot of focus is put on making the carbon spheres and the ZnO HS, although I believe these have been reported before. This should be clearly stated with proper referencing.

Respond: we fully accept your comments. In this revised manuscript, we have re-written the corresponding part in Introduction to highlights our work’s novelty comparing with previous reports.

Comment 2.       The manuscript contains a lot of mistakes and the English is not that good. Please carefully go through the text and figures again.

Respond: we sincerely apologize for our carelessness and fully accept your comments. We have carefully checked the manuscript, tried our best to eliminate language mistakes and typos in text and figures. All changes in the revised manuscript have been highlighted by “Track Changes” function in Microsoft Word.

Comment 3.       The authors claim that the presented solar cells are QD-sensitized solar cells (QDSSC). However, to me it is unclear if these cells can really be called QDSSC. This would require that the Zn-Cd-Se material consisted of nanometer-sized particles (which it might do). Figure 6 clearly highlights that the conversion affects the band gap of the HS. However, is it really a quantum effect or would you expect the same thing when converting bulk ZnO to bulk Zn-Cd-Se? Nonetheless, in the absence of a better name for this type of solar cell, QDSSC would be fine (although it should be discussed).

Respond: we thank for your comment. As you suggested, the Zn-Cd-Se was transformed from ZnO hollow sphere which was assembled nanometer-sized particles, however, this materials’ application in solar cell is based on the wording principle of quantum dots-sensitized solar cell and it seems that there is no better name for this type of solar cell, therefore, we believe that name this type of solar cell as QDSSC is proper now. A for the figure 6, the band gap variations among ZnO HS, ZnSe@ZnO HS, and ZnxCd1-xSe@ZnO HS were caused by the formation of new materials on ZnO HS surface.

Comment 4.       Step-wise ion exchange reactions are used to convert the ZnO to Zn-Cd-Se. The authors claim that this produces a double shell structure (ZnxCd1-xSe@ZnO). However, in my opinion, there is no evidence for this (e.g. EDS is not sensitive enough to claim this). I see no reason why not the entire ZnO structure would be converted to Zn-Cd-Se. Has this been tested by for instance a reaction time series? Furthermore, could the smaller spheres be more efficiently converted to Zn-Cd-Se compared to the large ones?

Respond: we thank you for your comment. In fact, the ion-exchange reactions did not produce double shell structure. During this process, the ZnxCd1-xSe was formed on surface of ZnO due to the ion-exchange between Se2- and O, Zn2+ and Cd2+. The ZnxCd1-xSe was directly grown on ZnO hollow sphere, it is still single shell after ion-exchange process. As you commented, we did not provide enough evidence to prove that the formation of ZnxCd1-xSe, therefore, in this revised manuscript, we provide an elemental mapping test result on surface of hollow spheres in figure S1 (Supplementary Materials), from which, Cd, Se, Zn, O elements can be scanned confirming the formation of ZnxCd1-xSe on surface of ZnO hollow spheres, together with EDS results from TEM image as shown in Fig. 4(f), we can prove that the formation of ZnxCd1-xSe on surface of ZnO HS.

  According to the changes, we have re-written the related part of our manuscript as follows:

Based on ZnO HS, the ZnxCd1-xSe@ZnO HS were further fabricated through ion-exchange process by soaking ZnO HS in Se2- and Cd2+ aqueous solution, respectively. The morphology variation has been recorded by SEM and TEM images. Fig. 4(a) and its inset show the surface morphology of ZnO HS, big particles structure can be observed. The fine structure is further revealed by TEM in Fig. 4 (b), the obvious shell and empty inside confirm the hollow spherical structure. The composition of hollow sphere is shown in Fig. 4(c), except Cu element from the carbon film supported copper grid, only Zn and O elements appear, and the atomic ratio is close to 1:1, confirming the formation of ZnO HS. After the ion-exchange process, a slight morphology variation on surface of hollow sphere is shown in Fig. 4(d) and its inset. Smaller nanoparticles are formed on surface, indicating new substance formed on surface of ZnO HS. In order to testify that the ZnxCd1-xSe can be formed on the surface of ZnO HS, the elemental mapping test results are shown in Figure S1 (see Supplementary Materials). As shown in Fig. S1, Zn, Cd, Se and O elements can be scanned on surface of hollow sphere, indicating that ZnxCd1-xSe has formed on surface of ZnO after ion-exchange process. Moreover, the TEM image in Fig. 4(e) shows an evident morphological variation after the ion-exchange process, also implying that some new materials formed on surface of ZnO HS. In addition, the EDS analysis of Fig. 4(f) from a designated inner spot of hollow sphere shell shows that after ion-exchange process, Cd and Se elements can also be found on the inner shell, and the atomic ratio of (Zn+Cd) to (Se+O) is close to 1:1, indicating the successful formation of ZnxCd1-xSe@ZnO HS.

Comment 5.       The authors claim that there are small particles inside the HS in the TEM image in Figure 4e. This is not necessarily true as the darker regions can also be from the top and bottom of the particle.

Respond: we accept your comment, and apologize for our inappropriate expression. In this revised manuscript we have re-written this par as follows:

Moreover, the TEM image in Fig. 4(e) shows an evident morphological variation after the ion-exchange process, also implying that some new materials formed on surface of ZnO HS.

6.       The authors make a long argument about how the “gradient structure” will improve the performance of the devices due to band edge realignment. In addition to the lack of evidence for a gradient structure (see point 4), there is also no evidence to claim this. Furthermore, according to the model described by the authors, the ZnO seems to be isolated inside the HS. This means that there would be no pathway of electrons throughout the photoanode. Thus, I think the argument in Figure 8d is quite much speculation (and perhaps not necessary).   

Respond: we accept your comment, and apologize for our carelessness on missing cite of references to support our argument. As Figure S1 and Figure 4 presented, we can prove that the ZnxCd1-xSe can form on outer and inner surface of ZnO by ion-exchange process. according to previous reports such as “J. Phys. Chem. C 2012, 116, 3802-3807” and “Nano Lett. 2011, 11, 4138-4143”, we believe that it is reasonable to preserve this argument, because that will help readers to understand the mechanism of ZnxCd1-xSe@ZnO HS in QDSSCs’ application. The missing references have been added in the revised manuscript to support our claim, which is shown as follows:

The ZnSe layer firstly formed ZnO HS surface during Se2- ion-exchange process, then a Cd2+ ion-exchange process would replace Zn2+ from surface of ZnSe layer to deep, forming sandwich composition gradient structure of ZnxCd1-xSe@ZnO HS with Cd2+ rich in surface [22, 33].

References

[22] Li, H.; Cheng, C.; Li, X.; Liu, J.; Guan, C.; Tay, Y.Y.; Fan, H.J. Composition-graded ZnxCd1–xSe@ZnO core-shell nanowire array electrodes for photoelectrochemical hydrogen generation, J. Phys. Chem. C 2012, 116, 3802-3807.

[33] Xu, J.; Yang, X.; Wang, H.; Chen, X.; Luan, C.; Xu, Z.; Lu, Z.; Roy, V.A.; Zhang, W.; Lee, C.S. Arrays of ZnO/ZnxCd1-xSe nanocables: band gap engineering and photovoltaic applications, Nano Lett. 2011, 11, 4138-4143.

7.       How do the current devices perform in comparison to other related devices in the literature?

Respond: we fully accept your comment. In this revised manuscript, we compared our device with similar QDSSC reported in literature, and the comparison result showed that our structure has better ability in improving the Jsc of solar cell, and leading to a better PCE than the compared devices.

The added discussion is shown in the revised manuscript as follows:

In comparison with previous report, our QDSSC’s Jsc and PCE outperform the similar ZnO-ZnxCd1-xSe QDSSC based on ZnO nanowire arrays (Jsc=6.7 mA cm-2, Voc= 0.64V, FF=0.35, PCE=1.5%) [32], indicating this kind of structure’s potential advantages in improving the Jsc of solar cells.

References

[32] Myung, Y.; Kang, J.H.; Choi, J.W.; Jang, D.M.; Park, J. Polytypic ZnCdSe shell layer on a ZnO nanowire array for enhanced solar cell efficiency, J. Mater. Chem. 2012, 22, 2157-2165.

Reviewer 2 Report

This manuscript proposes the synthesis of ZnxCd1-xSe@ZnO hollow spheres in different sizes for quantum dots sensitized solar cells application. The topic is interesting, and certainly consistent with the contents to be proposed to the readers of “Nanomaterials”. However, the manuscript is not so well written and should be improved to be read with pleasure: this represents an important aspect in the current scenario of publications in international journals. Overall, I think that this manuscript could be accepted if the Authors will be able to take into account the following major revisions (in terms of bibliographic updates, grammar corrections and content deepening):

-          Detailed revisions: I spent several hours reading this manuscript, and Authors are asked to follow carefully the attached PDF file where I highlighted some points to be addressed. The attached file also contains language mistakes and typos (they are many in this work and should not be present when submitting a manuscript to an international journal: Authors are asked to check the manuscript better next time); some questions related to manuscript contents could also be present and Authors must consider them properly before submitting the revised manuscript. A point-by-point reply is required when the revised files are submitted.

-          Considering the amount of mistakes and typos present in this manuscript, a further check carried out by a native English speaker or by a professional English language center is suggested.

-          The Introduction should give a wider overview on the present scenario related to DSSCs, both in terms of recently published reviews and research articles. In particular, emergent concepts in the field of electrolytes and flexible architectures are missing and a paragraph on this topic is highly suggested to be added in the Introduction. Authors are invited to go through the literature published in the last six months on these issues, and also on concepts developed some years ago in this field. Some of them are also mentioned in the above mentioned PDF file.

-          Authors should provide a clear explanation on the experimental error of the proposed research work. In particular, reproducibility of the phenomena described in the manuscript should be clearly stated in the “Results and Discussion” section; besides, some notes in the “Materials and Methods” section should be added highlighting which kind of experimental approach has been followed to check the reproducibility of the proposed system, the latter being of noteworthy importance in the present research field.

Author Response

Respond to Reviewer #2

Comments and Suggestions for Authors

This manuscript proposes the synthesis of ZnxCd1-xSe@ZnO hollow spheres in different sizes for quantum dots sensitized solar cells application. The topic is interesting, and certainly consistent with the contents to be proposed to the readers of “Nanomaterials”. However, the manuscript is not so well written and should be improved to be read with pleasure: this represents an important aspect in the current scenario of publications in international journals. Overall, I think that this manuscript could be accepted if the Authors will be able to take into account the following major revisions (in terms of bibliographic updates, grammar corrections and content deepening):

Re: Reviewer #2

We thank for your supporting to publication of our manuscript and appreciated your valuable comments. These comments helped us a lot to improve our manuscript. We sincerely accept your suggestions and have carefully revised the manuscript based on your comments, and made a list of responses to these comments. The changes have been highlighted by “Track Changes” function in Microsoft Word.

 Comment 1. Detailed revisions: I spent several hours reading this manuscript, and Authors are asked to follow carefully the attached PDF file where I highlighted some points to be addressed. The attached file also contains language mistakes and typos (they are many in this work and should not be present when submitting a manuscript to an international journal: Authors are asked to check the manuscript better next time); some questions related to manuscript contents could also be present and Authors must consider them properly before submitting the revised manuscript. A point-by-point reply is required when the revised files are submitted.

Respond: we thank you for your careful reading our manuscript and pointing out language mistakes and typos. We sincerely apologize for our carelessness and accept your comments. In this revised round, we have carefully checked the manuscript, tried our best to eliminate language mistakes and typos. As for the other questions related to manuscript contents, we have also reorganized according to your comments. All changes in the revised manuscript have been highlighted by “Track Changes” function in Microsoft Word.

 Comment 2. Considering the amount of mistakes and typos present in this manuscript, a further check carried out by a native English speaker or by a professional English language center is suggested.

Respond: We fully accept your comment and sincerely apologize for our bad English expression. In this revised manuscript, the English expression has been carefully checked and revised to improve the language by an invited native English speaker.

Comment 3. The Introduction should give a wider overview on the present scenario related to DSSCs, both in terms of recently published reviews and research articles. In particular, emergent concepts in the field of electrolytes and flexible architectures are missing and a paragraph on this topic is highly suggested to be added in the Introduction. Authors are invited to go through the literature published in the last six months on these issues, and also on concepts developed some years ago in this field. Some of them are also mentioned in the above mentioned PDF file.

Respond: we accept your comment and thank you for your recommendation of literatures. These recommended literatures on DSSCs enlarged our visions and gave insights on DSSCs fields. In this revised manuscript, we added the introduction on field of electrolyte and flexible architectures and cited the recommended literatures in the “Introduction” part.

 Comment 4. Authors should provide a clear explanation on the experimental error of the proposed research work. In particular, reproducibility of the phenomena described in the manuscript should be clearly stated in the “Results and Discussion” section; besides, some notes in the “Materials and Methods” section should be added highlighting which kind of experimental approach has been followed to check the reproducibility of the proposed system, the latter being of noteworthy importance in the present research field.

Respond: we fully accept your comment. In this revised version, we provide a figure S2 in Supplementary Materials to reveal the reproducibility of the performance of our QDSSCs based on ZnO HS with different sizes from 300 nm to 800 nm by repeatedly test the three QDSSCs prepared under same conditions. The re-written part was shown in revised manuscript as follows:

As the I-V curves show in Fig. 10(a), the mixture of hollow spheres in different sizes can further improve the Jsc to 20.77 mA cm-2, leading to the enhancement of PCE to 2.95%. To guarantee the performance’s reproducibility of our QDSSC based on ZnO HS with different sizes from 300 nm to 800 nm, we also prepared three QDSSCs under the same conditions for I-V test to investigate the stability of the photoanodes. The compared I-V test results are shown by Figure S2 in Supplementary Materials, which shows that a relative stable variation PCE performance (from 3.05% to 2.92%).

Round  2

Reviewer 2 Report

The manuscript has been properly revised and I recommend its publication.